# Efficient Bayesian Experiment Design with Equivariant Networks

**Conor Igoe***
Machine Learning Department
Carnegie Mellon University
cigoe@cs.cmu.edu

**Tejus Gupta***
Robotics Institute
Carnegie Mellon University
tejusg@cs.cmu.edu

**Jeff Schneider**
Robotics Institute
Carnegie Mellon University
schneide@cs.cmu.edu

## Abstract

Recent work in Bayesian Experiment Design (BED) has shown the value of using Deep Learning (DL) to obtain highly efficient adaptive experiment designs. In this paper, we argue that a central bottleneck of DL training for BED is belief explosion. Specifically, as an agent progresses deeper into an experiment, the effective number of realisable beliefs grows enormously, placing significant sampling burdens on offline training schemes in an effort to gather experience from all regions of belief space. We argue that choosing an appropriate inductive bias for actor/critic networks is a critical component in mitigating the effects of belief explosion and has so far been overlooked in the BED literature. We show how Graph Neural Networks are particularly well-suited for BED DL training due to their domain permutation equivariance properties, resulting in multiple orders of magnitude improvement to sample efficiency compared to naive parameterizations.

## 1 Introduction

Scientific inquiry depends on empirical observations to refine our understanding of the world. These observations generate data that can be analyzed, interpreted, and used to test hypotheses and theories. However, conducting experiments comes with costs, including time, financial resources, and logistical constraints. As a result, determining which observations are most valuable for a given line of inquiry is a central concern in *the Design of Experiments*.

In particular, Bayesian Experiment Design, or BED, has emerged as an elegant formalism for understanding the value of different experiment designs [1, 2]. Moreover, in recent years there has been a growing interest in adopting Deep Learning (DL) and Deep Reinforcement Learning (DRL) techniques to obtain effective experiment designs for BED tasks [3, 4, 5, 6, 7, 8, 9, 10].

Principal among the motivations for the involvement of DL techniques is their potential to increase the scope of problems that admit practical BED solutions [11]. More specifically, practitioners typically trade off decision-theoretic performance with online computational costs when deploying BED policies in real-world tasks. Deep Learning can amortize computationally costly but high-performing BED policies, allowing for improved decision-making, reduced compute burdens, or both. For example, recent work has shown that multistep lookahead-based approaches for achieving non-myopic experiment designs involve considerable online computation in the face of nested integrals. Wu and Frazier [12] describe how these approaches can require up to an hour of compute for a single decision in modest Bayesian Optimization (BO) problems. Although they propose a novel method to reduce this compute overhead to at most several minutes, such online compute requirements are prohibitive in certain high-frequency or resource-constrained tasks, for example in remote

---

*Equal contribution.

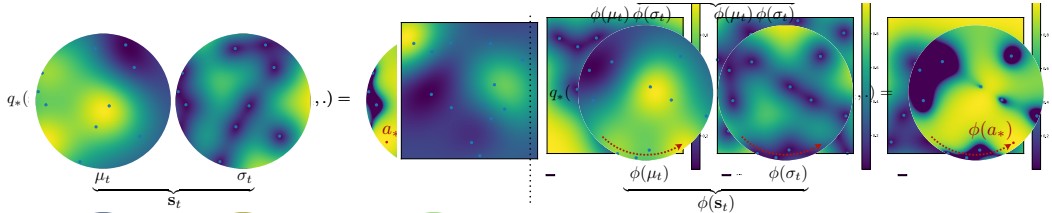

Figure 1: **Illustration of Domain Transformation Equivariance** The figure on the left of the dotted line shows the posterior mean, $\mu$, and marginal standard deviation, $\sigma$, of a 2D Gaussian Process, along with the LogEI acquisition function used in Bayesian Optimization. This corresponds to the optimal 1-step lookahead $q_*$ landscape when cast as an MDP, with the optimal action $a_*$ shown in red. The figure on the right shows the same posterior belief state and acquisition function under a transformation $\phi$ that rotates the domain by $90°$. Note that although these belief-action pairs appear different, they are related by the transformation $\phi$ and represent equivalent decisions under an equivariant policy.

sensing [10] or edge computing [13]. In contrast, amortized policies trained using DRL can achieve non-myopic designs while requiring several orders of magnitude less online computation [4, 10].

Although the recent focus on amortizing expert policies for BED has shown great promise, a core argument of this paper is that existing approaches have significantly high training compute burdens. For example, we find that these approaches can take on the order of days to match the performance of simple experts on modest BED tasks.

We trace this issue to an explosion of possible posterior beliefs as inference progresses through an adaptive experiment, which we call "belief explosion". We observe that existing approaches struggle to learn in the presence of this large diversity of beliefs at deeper timesteps. We argue that designing methods that can cope with this belief explosion efficiently should be a priority for the BED community, and our work proposes to leverage the equivariant structures of optimal BED policies to achieve this goal.

Specifically, we show how a large family of BED tasks admit optimal policies that are *domain permutation equivariant*, and that exploiting this structure leads to significantly more efficient learning in the face of belief explosion. Moreover, as we show in this work, these structural properties naturally suggest an inductive bias that is readily captured in Graph Neural Network (GNN) architectures.

In particular, our contributions are as follows:

- we show that a significant bottleneck when training policies for BED tasks is belief explosion, and show how standard architectures commonly found in the literature are ill-suited to address this challenge;

- we prove that a large family of BED tasks are domain permutation equivariant, which suggests a natural inductive bias for policy and critic networks;

- we demonstrate the utility of leveraging this domain-permutation equivariance on two prominent BED tasks, showing up to two orders of magnitude improvement in sample efficiency compared to various architectures operating on both belief-state and information-set representations;

- we show that GNNs trained on small BED tasks can be used to generalise to significantly larger tasks, resulting in order of magnitude less online compute at test time;

- we show that these equivariances can be extended to information-set-based policies in continuous BED tasks, allowing us to efficiently train performant policies for higher dimensional BO tasks.

## 2  Notation & Preliminaries

In order to provide a unified notation for our main theoretical and experimental results, we introduce the Bayesian Experiment Design and Markov Decision Process formalisms in the following two subsections, followed by a short overview of Graph Neural Networks.

## 2.1 Bayesian Experiment Design

Bayesian Experiment Design is a general framework that helps agents make intelligent decisions to efficiently reduce their uncertainty over a set of hypotheses. More specifically, BED models a sequential interaction process, or "experiment", between a decision-making agent and an environment containing some latent variable of interest, denoted $\theta$. A BED task is defined by the tuple $(\Theta, \mathcal{X}, \mathcal{Y}, p_0, p, r)$, where $\Theta$ defines a space of possible values for the latent variable, or "hypotheses", $\mathcal{X}$ defines a space of actions available to the agent, or "experiment designs", and $\mathcal{Y}$ defines a space of observations, or "experiment outcomes". The probabilistic models $p_0(\theta)$ and $p(y|x,\theta)$ define a prior over the possible hypotheses and the observation model, respectively.

At each time step $t = 0, 1, \ldots$, the agent chooses the experiment design $x_t$ based on the posterior belief state $p_t := p(\theta | \{(x_\tau, y_\tau)\}_{\tau=0}^{t-1})$. The environment then yields the observation $y_t \sim p(y|x_t, \theta)$, which the agent adds to its information set $\mathcal{I}_t = \{(x_\tau, y_\tau)\}_{\tau=0}^{t-1}$. The agent is interested in choosing informative experiment designs to reduce the uncertainty about $\theta$ as measured by some functional of the posterior $r(p(\theta|\mathcal{I}_t))$.

The BED framework is flexible in its ability to model finite hypothesis classes as well as continuous. For example, in this paper we focus on two tasks that feature prominently in the ML and robotics literature as case studies for our experimental results: Bayesian Optimization (BO) and Active Search (AS). BO can be thought of as an instance of a BED task where $\theta$ represents some unknown continuous black-box function that must be optimized. Similarly, AS can be thought of as a BED task where $\theta$ is a binary-valued vector representing the location of objects of interest in a discredited real-world terrain. We return to more detailed descriptions of the BO and AS tasks in their respective experimental sections.

## 2.2 Markov Decision Processes

For ease of analysis, we now describe how to view a BED task as an instance of a Markov Decision Process (MDP).

We consider finite-horizon MDPs defined by the tuple $(\mathcal{S}, \mathcal{A}, P, \rho, r, T)$, where the state space $\mathcal{S}$ is continuous and the action space $\mathcal{A}$ is discrete, and the unknown state transition function $P : \mathcal{S} \times \mathcal{S} \times \mathcal{A} \to [0, \infty)$ represents the probability density of the next state $\mathbf{s}_{t+1} \in \mathcal{S}$ given the current state $\mathbf{s}_t \in \mathcal{S}$ and action $\mathbf{a}_t \in \mathcal{A}$. The environment generates a reward $r_t \in \mathbb{R}$ based on the current state $\mathbf{s}_t$ and action $\mathbf{a}_t$. The environment generates trajectories of length $T \in \mathbb{N}$ starting from an initial state drawn from initial state distribution $\rho : \mathcal{S} \to [0, \infty)$. The standard MDP objective is to find a policy $\pi_\varphi : \mathcal{S} \to \mathcal{A}$ by optimizing policy parameters $\varphi$ to maximize the expected discounted return $\mathbb{E}[\sum_{t=0}^{T-1} r_t | \pi_\varphi]$. Note that we use $t$ to denote the MDP timestep throughout this paper.

For the BED task, we define the MDP state as the posterior belief $\mathbf{s}_t := p_t$, and the action as the design $\mathbf{a}_t := x_t$. The state definition satisfies Markovian dynamics assumptions by stochastically transitioning to state $p_{t+1}$ after incorporating $(x_t, y_t)$ using the inference equations, with stochasticity arising from the random variable $y_t$. The reward is defined as a functional of $p_t$, and can involve a variety of application-specific quantities that incorporate information-theoretic terms derived from the posterior belief state, as well as action costs and geometric quantities relating to the geometry of the underlying domain. In the experiment section, we describe some example reward functions that are commonly found in the BED and related literature for our case study environments.

We note that it is also possible to cast BED tasks in the MDP formalism by defining the state $\mathbf{s}_t := \mathcal{I}_t$. The advantage of choosing information set state representations over belief-state representations is that it allows for the training of end-to-end networks that amortize over the inference process mapping from $\mathcal{I}_t$ to $p_t$, as well as the expected returns and optimal action. This may be attractive in certain contexts, where inference itself is costly. Indeed, this is the approach adopted in [3, 8, 14, 15, 8]. Due to space constraints, in the main paper we present results that leverage discretized belief state representations following previous work [7, 10, 16]. We include additional results on non-discretized information set methods in the Appendix E.2.

## 2.3 Graph Neural Networks

Graph Neural Networks (GNNs) are a broad family of architectures designed to process data represented as graphs, where entities are modeled as nodes and their interactions as edges. Through iterative message passing, GNNs enable each node to update its representation by aggregating information from its neighbors, thereby capturing relational structure and inductive biases that standard feedforward or convolutional architectures cannot. This makes them especially well-suited to domains where the underlying structure is non-Euclidean or relational.

Over the past several years, GNNs have been successfully applied across a wide range of machine learning and robotics subfields. In representation learning, GNNs underpin molecular property prediction [17], and structure-aware scene understanding [18]. In reinforcement learning and control, GNNs have been used to model multi-agent interactions [19], physical dynamics [20], and robotic manipulation [21], leveraging their ability to encode symmetry and compositional structure. More broadly, GNNs have become a unifying framework for exploiting invariances and equivariances inherent in structured decision-making problems.

In our experiments, we build on these insights by leveraging GNNs to capture important equivariances in Bayesian Experiment Design (BED) tasks. As described in Section 4, our model architecture exploits the graph structure of the belief representation and design space to improve generalization across environments and hypothesis classes.

## 3  Belief Explosion as a Policy Optimization Bottleneck in BED

In this section, we identify the *belief explosion* issue and identify how it affects policy optimization for BED tasks. More specifically, belief explosion refers to the phenomenon in which the number of possible beliefs an agent must consider becomes intractably large as the search progresses. This is a well-known problem in the POMDP planning literature [22, 23], but has received little attention in the contemporary BED and deep learning communities.

Figure 2 visualizes the belief explosion for a 1D BO task. The figure on the left shows four belief state trajectories, illustrating the increasing diversity of beliefs with time. The figure on the right illustrates this more precisely: As the agent progresses further into the trial, the average divergence between two realizable beliefs increases. This is shown by the blue curve, which measures the KL divergence between canonical representations of the beliefs—the posterior mean vector and covariance matrix—in discretized Gaussian Process prior BO tasks. The orange curve shows that much of this growth is overstated: for each pair of realizable beliefs $(\mathbf{s}, \tilde{\mathbf{s}})$, there exists a transformed belief in the set of all permuted beliefs $\mathcal{B}(\tilde{\mathbf{s}})$ which can have significantly lower divergence with $\mathbf{s}$ compared to its original untransformed representation.

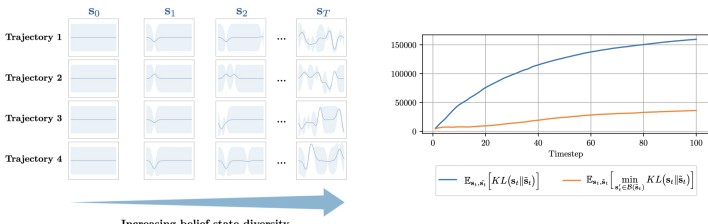

Figure 2: **Belief Explosion**

Next, we show that this rapid increase in $\mathcal{S}_t$ with $t$ renders current policy optimization methods for BED highly sample inefficient when using standard belief representations and network architecture choices. We consider a simple offline learning problem in a small 10-bin discretized 1D BO task. The goal here is to imitate the 1-step greedy expected information gain expert using behavior cloning with a fully connected network trained on $\texttt{concatenate}(\mu_t, \texttt{flatten}(\Sigma_t))$ discretized belief-state representations. Figure 3 shows cross entropy train-test loss curves with an expert data set of $10^4$ expert environment transition samples.

Mild overfitting is evident in the left plot of Figure 3 and is perhaps otherwise unremarkable. The figure on the right in Figure 3 shows the same test loss curve but is decomposed over various timesteps. As an example, the green curve shows the test loss when using test beliefs $\{p_3^i\}$ as input to $\pi_\varphi$.

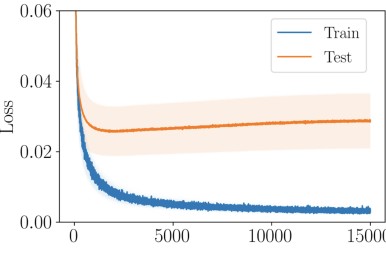 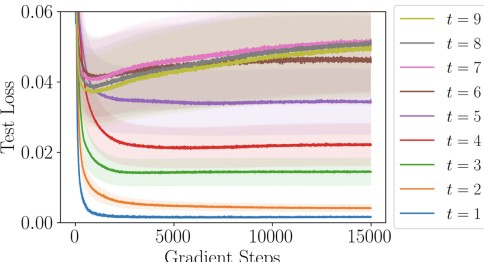

Figure 3: **Test Loss Temporal Decomposition** Note that each curve is averaged over 10 seeds. The shaded region shows $\pm 2$ standard errors.

We make the following observations about Figure 3:

1. The best test loss achieved by $\pi_\varphi$ increases monotonically as a function of $t$ until timestep $t = 6$, after which the best test loss stagnates. That is, for "shallow" beliefs, $\pi_\varphi$ generalizes well, while for "deep" beliefs, generalization performance plummets. Given that each timestep is associated with the same number of training samples, as is common in standard rollout-based DRL data collection strategies, Figure 3 is a clear illustration of belief explosion in full effect;

2. While the aggregate test curve in left plot of Figure 3 shows mild overfitting, decomposing the test loss over the timesteps tells a very different story: namely, that for shallow beliefs, our network hasn't yet converged to the best test loss, while for deep beliefs, we experience drastic overfitting.

The key issue is that a fully connected network fails to incorporate the structural properties of the BED task, and as a result, it struggles to generalize to unseen belief states at test time. This limitation poses challenges even in very small BO instances, such as this task with only 10 discrete bins.

We argue that embedding BED-specific inductive biases into the model architecture is essential for efficient learning. Our key insight is that naive network parameterizations experience belief explosion along the blue curve of Figure 2, while networks that leverage a property we call *domain permutation equivariance* (introduced in the next section) experience a significantly tamer growth of belief diversity according to the orange curve.

## 4 Domain-permutation equivariance in BED and GNNs

In this section, we present some intuition about the structural properties of BED tasks that can be exploited for efficient learning. Figure 1 illustrates how we might intuitively expect the acquisition values of belief $p_t$ to be equivariant with respect to a simple transformation of the domain $\mathcal{X}$. We formalize this intuition in our main theoretical result below about the domain-permutation equivariance of optimal policies in BED tasks and use it to motivate the use of GNNs as an appropriate inductive bias.

Specifically, for ease of exposition, we focus our attention on BED tasks in which the posterior belief over the unknown function is fully characterized by the first two moments, as is the case in Bayesian Optimization and Active Search.

Let $f : \mathbb{R}^d \to \mathbb{R}$ be the unknown function which the active sensing agent is querying. We consider BED tasks over a discretised space $\mathcal{X}_{\text{disc}} = \{x^1, x^2, \ldots, x^M\}$. The posterior distribution over $f$ (belief-state) given the information set $\mathcal{I}_t$ is therefore a tuple of finite dimensional moments $\mathbf{s}_t = (\mu_t, \Sigma_t)$ with:

$$\mu_t = \left(\mathbb{E}[f(x^i)|\mathcal{I}_t]\right)_{i=1}^M \in \mathbb{R}^M \text{ and } \Sigma_t = \left(\mathbb{E}[(f(x^i) - \mu_t^i)(f(x^j) - \mu_t^j)|\mathcal{I}_t]\right)_{i,j=1}^M \in \mathbb{S}_+^M.$$

Consider a set of permutations $\Phi$, where each permutation $\phi \in \Phi$ permutes the belief state $\mathbf{s}_t$. More specifically, each permutation is a bijection $\phi : \{1, 2, \ldots, M\} \to \{1, 2, \ldots, M\}$ that reorders the indices of the $M$ input points in $\mathcal{X}_{\text{disc}}$. Thus, $\phi(\mathbf{s}_t) = (\mu_{\phi,t}, \Sigma_{\phi,t})$ where:

$$\mu_{\phi,t} = \left(\mathbb{E}[f(x^{\phi(i)})|\mathcal{I}_t]\right)_{i=1}^M = \left(\mu_t^{\phi(i)}\right)_{i=1}^M,$$

and
$$\Sigma_{\phi,t} = \left( \mathbb{E}\left[ \left( f(x^{\phi(i)}) - \mu_t^{\phi(i)} \right) \times \left( f(x^{\phi(j)}) - \mu_t^{\phi(j)} \right) \big| \mathcal{I}_t \right] \right)_{i,j=1}^{M} = \left( \Sigma_t^{\phi(i),\phi(j)} \right)_{i,j=1}^{M}.$$

We overload the notation $\phi$ to also map actions to permuted actions. In our active sensing setup, each action $\mathbf{a}$ senses some subset of $\mathcal{X}_{\text{disc}}$, i.e., $\mathbf{a} = \mathcal{X}_{\mathbf{a}} \subseteq \mathcal{X}_{\text{disc}}$, and we define $\phi(\mathbf{a}) = \{x^{\phi(i)} : x^i \in \mathcal{X}_{\mathbf{a}}\}$.

**Theorem 1** (**Domain Permutation Equivariance**). *Let $\Phi$ be a set of permutations. Suppose a Bayesian Experiment Design (BED) task satisfies, for all $\phi \in \Phi$ and all state-action-next-state transitions $\mathbf{s}, \mathbf{s}' \in \mathcal{S}, \mathbf{a} \in \mathcal{A}$:*

*(i) **Reward invariance:*** $\quad r(\mathbf{s}, \mathbf{a}) = r(\phi(\mathbf{s}), \phi(\mathbf{a}))$

*(ii) **Transition invariance:*** $\quad p(\mathbf{s}' \mid \mathbf{s}, \mathbf{a}) = p(\phi(\mathbf{s}') \mid \phi(\mathbf{s}), \phi(\mathbf{a}))$

*Then, for all $\phi \in \Phi$, all $\mathbf{s} \in \mathcal{S}, \mathbf{a} \in \mathcal{A}$, and $\gamma \in [0,1]$:*

$$\pi_\gamma^*(\phi(\mathbf{s})) = \phi(\pi_\gamma^*(\mathbf{s})) \quad \text{(optimal policy equivariance)}$$
$$Q_\gamma^*(\phi(\mathbf{s}), \phi(\mathbf{a})) = Q_\gamma^*(\mathbf{s}, \mathbf{a}) \quad \text{(optimal Q-value invariance)}.$$

This reward and transition equivariance property holds for a wide range of BED tasks. For example, with the BO task, a practitioner might choose the reward as the negative posterior argmax entropy. Since the transition is the GP inference, both reward and transition invariance hold, since neither depend explicitly on the ordering of states in the domain. These conditions also hold for the AS task with a more constrained set of permutations, which is described in detail in the appendix.

One way to interpret this result is that using mean *vectors* and covariance *matrices* as belief representations during learning implicitly imposes a fixed ordering on the underlying random variables. This is problematic, as the BED task ultimately requires reasoning about *sets* of jointly distributed random variables, which are inherently unordered. By designing policies that are equivariant to permutations of these variables, we enable agents to generalize optimal behaviors across a broader class of decision-theoretically equivalent belief states, thereby substantially improving sample efficiency.

We emphasize that this theorem is complementary to Theorem 3 by Foster et al. in [3], which concerns *observation-set permutations*. Whereas observation-set permutation invariance focuses on temporal structure, *domain-permutation equivariance* is concerned with the geometric structure of the belief state itself.

The significance of this result is that we can easily generalize between permuted belief states during policy optimization. We use a straightforward way of doing so: using graph representations of the posterior belief as our state, with Graph Neural Networks as our policy class, which are permutation equivariant by design.

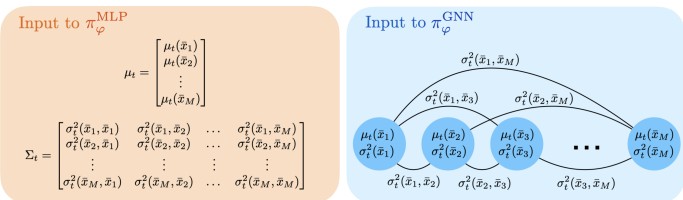

Figure 4: **Dense Network & GNN Comparison: Input Representations** Note that although the graph input to $\pi_\varphi^{\text{GNN}}$ is a complete graph, we omit all edges from the figure for ease of illustration.

We use a complete undirected graph to represent our belief, with $M$ nodes representing the unknown function value at points in the discretized domain $\mathcal{X}_{\text{disc}}$, and $M(M-1)/2$ edges representing our joint uncertainty between points in $\mathcal{X}_{\text{disc}}$. For example, in the Bayesian Optimisation setting, each node in the graph is labeled with a 2d vector containing posterior statistics $\left( \mu_t(x) = \mathbb{E}(f(x)|\mathcal{I}_t), \, \sigma_t^2(x) = \mathbb{V}(f(x)|\mathcal{I}_t) \right) \in \mathbb{R}^2$. Each edge of the graph is labeled with the posterior covariance $\mathbb{C}\text{ov}\left( f(x), f(x') | \mathcal{I}_t \right) \in \mathbb{R}^+$. We then use a `TransformerConv` architecture [24] for the GNN which transforms labels on the nodes by combining information from the labels in neighboring nodes and edges using learned soft attention. Figure 4 illustrates the input representation to the GNN and the MLP.

# 5 Experiments

Our experiments aim to answer the following questions:

1. Do GNNs offer an effective inductive bias for efficiently learning BED policies using data from an expensive oracle?
2. Can GNNs be used to learn non-myopic BED policies using reinforcement learning?
3. How well do GNN-based policies transfer across BED problems, and do they scale to larger search problems?

Note that due to space constraints, we leave our reinforcement learning, generalization, and continuous BO experiments in Appendix E.2.

## 5.1 Setup

**Tasks:** We use two BED tasks: Bayesian Optimization (BO) and Active Search (AS).

**Bayesian Optimization** models the problem of efficiently locating the argmax of an unknown function using black-box queries. Our latent variable of interest $\theta$ is an unknown function $f : \mathcal{X} \to \mathbb{R}$ defined on the domain $\mathcal{X} \subset \mathbb{R}^d$. The action space $\mathcal{A}$ consists of point locations in the domain $x \in \mathcal{X}$, and the observation model $p(y|x, \theta)$ is a black-box evaluation at the detection location with Gaussian noise $p_{\text{noise}} = \mathcal{N}(0, \sigma_{\text{noise}}^2)$. The prior $p(\theta)$ is defined using a zero-mean Gaussian prior with a chosen kernel.

For our experiments, we use a Squared-Exponential kernel $k(x, x') = \sigma^2 \exp\left(-\frac{\|x-x'\|_2^2}{2\ell^2}\right)$. We also discretize the domain $\mathcal{X}$ into a finite set of points $\{x^i\}_{i=1}^M = \mathcal{X}_{\text{disc}} \subset \mathcal{X}$, allowing for exact closed-form inference over the set of random variables $\{f(x^i)\}_{i=1}^M$ using standard Gaussian inference equations [25]. We also restrict the agent's actions to $\mathcal{X}_{\text{disc}}$.

There are several candidate choices for the MDP reward function, but a natural choice [26, 27] for BO tasks is to define $r_t = -\mathcal{H}(p_t^*)$. Here we define $p_t^*$ as the posterior over the $\arg\max$ of $f$, and we use $\mathcal{H}$ to denote the Shannon entropy. With this reward definition, we incentivize designs that improve the agent's understanding of the location of the maximum of $f$, or equivalently, designs that reduce the agent's posterior uncertainty about the location of an optimal function value. We note that, unlike posterior inference, there are no closed-form expressions available for $\mathcal{H}(p_t^*)$. We therefore use Monte Carlo sampling to approximate $p_t(x^*)$ with the following estimator:

$$\hat{p}_t(x^*) = \frac{1}{N} \sum_{i=1}^N \mathbb{I}\left[\arg\max_x f_i(x) = x^*\right]$$

where $f_i \sim p_t(f)$ and then use $r_t = -\mathcal{H}\left(\hat{p}_t(x^*)\right)$.

We use a 1D domain with $M = 32$ grid points in region length $= 8$ with $\ell = 1$, $\sigma_{\text{noise}}^2 = 0.1$ and $\sigma^2 = 1$. These parameters were chosen such that draws $f \sim p_0$ typically have 3-4 local optima, and the smoothness is qualitatively discernible. We set the episode length to $T = 8$. As we shall see, although this toy problem instance may seem trivial, it poses significant challenges when training with standard dense neural networks. In the appendix we explore fully continuous non-discretized BO tasks.

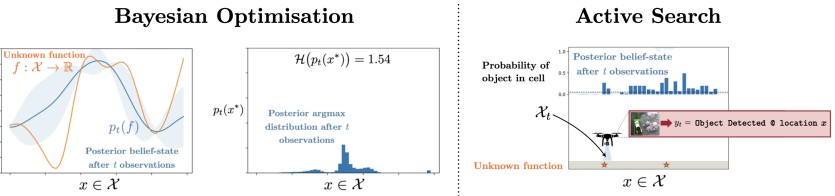

Figure 5: **Overview of the Bayesian Optimization and Active Search Tasks**

**Active Search** models the problem of efficiently locating sparse objects in an unknown environment by actively taking sensing actions given all observations thus far [28, 29, 30, 31, 32]. Figure 5 illustrates an Active Search problem using an aerial remote sensing agent. We adopt the environment model from [10].

The Active Search problem can be thought of as an instance of the standard Bayesian Optimization problem with some modifications, the key differences being:

1. BO: $f$ smooth; AS: $f$ sparse

2. BO: pointwise sensing of $f$; AS: contiguous interval sensing of $f$

3. BO: homoskedastic noise; AS: heteroskedastic noise based on sensed interval $\mathcal{X}_t$.

Here, $f : \mathcal{X} \to \{0, 1\}$ denotes object presence, with $\mathcal{X}$ representing a 1D, 2D, or 3D search space. The prior $p(\theta)$ assumes sparsity. Instead of selecting single points, the agent senses contiguous intervals $\mathcal{X}_t \subseteq \mathcal{X}$, receiving a noisy observation $f(\mathcal{X}_t)$. Wider intervals yield more noise, reflecting the trade-off between resolution and coverage (e.g., higher altitudes capture more area but with less precision).

We adopt the recovery reward, which selects actions to maximize expected detections under the posterior [33, 10, 34].

Experiments use a 1D grid with $M = 32$ and episode length $T = 32$. Actions span up to $l = 4$ points, with noise increasing with interval size—inducing a trade-off between exploration and precision.

**Learning-based baselines:** We compare GNNs against three standard parameterizations of the policy and critic networks used in previous work: FCNs, CNNs, and Transformers on the information set.

FCN or Dense networks are a naive, but common choice in the DRL literature [35, 36]. Multiple BED DRL papers have used similar architectures while focusing primarily on other dimensions of the BED problem [7, 10].

CNNs, on the other hand, provide a stronger inductive bias for BED tasks [7]. They are set up to predict the Q value or the action probability for each point in the domain using marginal statistics at that point. Although this architecture can be efficient at learning certain functions, it completely ignores the correlations between different points in the domain and, therefore, is insufficient for representing the multistep value function or optimal policy for BED tasks.

Transformers are another promising alternative [37], capable of processing entire information sets directly. Though there is some similarity in transformers and GNNs, they operate on different state representation and offer distinct invariances: transformers directly operate on the information set and provide information set permutation invariance, while the GNNs operate on the belief state and additionally provide domain permutation-set equivariance. Note that in the appendix we include more experiments comparing Transformers and information-set-based GNNs without requiring any discretization strategies.

**Non-amortized baselines:** For Bayesian Optimization, we compare our learned policies with the greedy policies w.r.t. the Expected Improvement [38] and UCB [39] acquisition functions, as well as Thompson Sampling [40]. For Active Search, we compare with random waypoint selection and a linear sweeping of the search domain [28].

## 5.2 Behavior Cloning Experiments

We begin our experiments by comparing different policy architectures for behavior cloning an expensive oracle and validating that GNNs offer a helpful inductive bias.

Our goal is to train policies to clone the 1-step greedy expert $\pi_{1\text{-step}} = \arg\max_a \mathbb{E}[r_{t+1} | p_t, a_t = a]$, with a negative posterior argmax entropy reward function. We use Monte Carlo rollouts to generate an offline dataset of beliefs encountered by the 1-step greedy expert, along with the expert actions: $\mathcal{D} = \{(p_t^i, a_t^i)\}$ where $p_t^i$ denotes the posterior belief state at timestep $t$ in trial $i$ when following policy $\pi_{1\text{-step}}$. In other words, $\mathcal{D}$ is a dataset of (state, actions-of-1-step-expert) collected from the 1-step lookahead policy. The Active Search task uses 32 timesteps per rollout, and the Bayesian

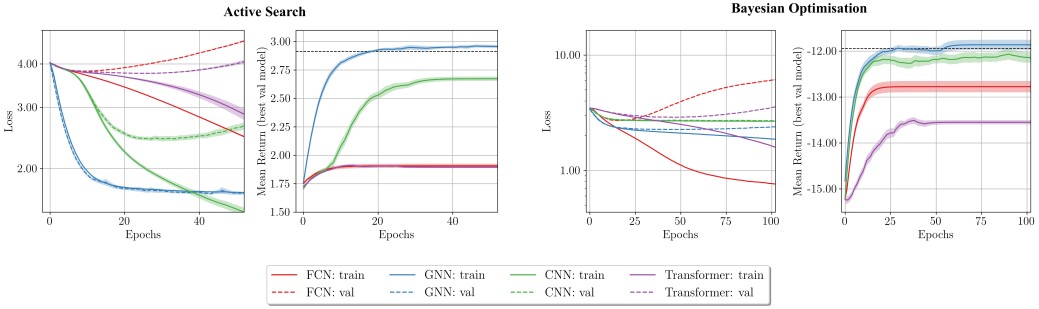

Figure 6: **Comparison of Inductive Biases for Behavior Cloning at 5,000 samples:** Training (solid) and test (dotted) behavior-cloning loss, and expected return for various networks on the Active Search (left) and Bayesian Optimization (right) tasks. The dotted black line denotes the oracle's expected return. During training, we keep track of the best-performing policy on the validation set using the behavior cloning loss (i.e., the rolling best policy). The expected return is then computed by evaluating this best-val policy. Note that each curve is averaged over 5 seeds. The shaded region shows $\pm 2$ standard errors.

Optimisation task uses 8. We note that we include behavior cloning results on BO instances for higher dimensional continuous BO tasks using 32 timesteps in the appendix.

Figure 6 shows the standard train-test loss curves and online performance for different policy networks trained using behavior cloning on AS and BO respectively. We train these networks on an 80:20 train:test split of data from $\mathcal{D}$ using Adam, with varying dataset size $|\mathcal{D}| \in \{50, 500, 5000, 50000\}$ (full results presented in the appendix). The dotted black lines on the right subfigures show the online performance of $\pi_{\text{1-step}}$.

Across both tasks and all dataset sizes, we observe that the GNN shows excellent generalization on the test set, while the other networks overfit and perform poorly on the test set. Moreover, the GNN is able to match the expert's rollout performance using 5,000 samples on both tasks.

Looking at the baselines, we see that the FCN and Transformer are very data-hungry and require a prohibitive number of samples even for small tasks. CNNs improve upon FCNs and Transformers and can learn much more efficiently. However, they are limited by their inability to use cross-correlation terms in the belief state, which is essential for doing well on many BED tasks.

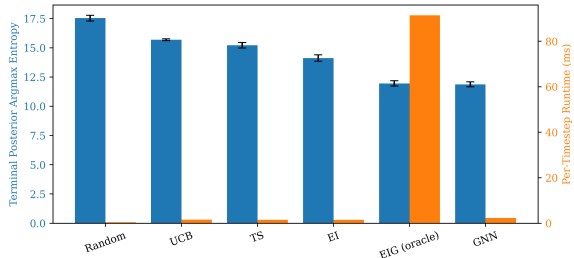

Figure 7: **Comparison with Non-Amortized baselines on BO**

Figure 7 compares the trained GNN with non-learning based baselines: Upper Confidence Bound (UCB), Thompsons Sampling (TS), Expected Improvement (EI) and Expected Information Gain (EIG). We report inference time and performance, as measured by the terminal posterior argmax entropy, that is, the entropy of the posterior argmax implied by the belief state at the final timestep of the rollout ($t = T = 8$). We see that the GNN can match the oracle's performance while being significantly faster. Acquisition functions such as EI and UCB are faster to compute, but don't perform as well.

Our experiments in this section validate our claim that the choice of inductive bias is a bottleneck for efficiently learning BED policies. We show that GNNs are more sample efficient than other networks by at least an order of magnitude on all tasks, confirming the benefit of exploiting the domain-permutation equivariance structure for BED tasks.

### 5.3 Reinforcement Learning Experiments

In this section we provide results on learning non-myopic policies using reinforcement learning. We use DDQN with nonstationary Q-values, integrated Bellman targets and various critic architectures to learn policies described in Section 2. As with the behavior cloning experiments, we compare the performance of the Graph Neural Networks with the Dense Networks, CNNs, and Transformers.

For the BO enviornment, we use the negative posterior argmax entropy as the reward function. For the AS environment, we use the full recovery rate reward [10].

Figure 8 shows the training curves for our reinforcement learning experiments. The black dotted line shows the performance of the greedy policy. As in the offline training setting, we see significant improvements in training efficiency when using GNNs. In fact, the GNNs are the only policy that can learn non-myopic policies in either task. All other networks train very slowly and we expect them to require at least an order of magnitude more training steps to even reach the myopic policy's performance.

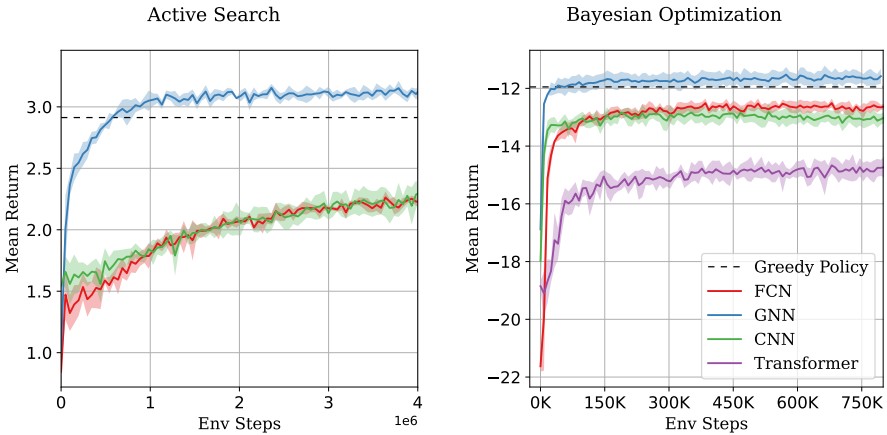

Figure 8: **Comparison of Inductive Biases for Reinforcement Learning:** RL training curves for various architecture on the AS (left) and BO (right) tasks. The dotted black line denotes the oracle's expected return. Note that each curve is averaged over 5 seeds. The shaded region shows $\pm 2$ standard errors.

## 6 Conclusion

In this work, we identified belief explosion as a bottleneck in policy optimization for BED tasks, and demonstrated that designing networks with the right equivariance structure is an effective strategy at taming this issue. We empirically show that our method can amortize expensive oracles, learn non-myopic policies, and generalize to larger scale BED tasks at test time. We also showed how GNNs can be leveraged in both discrete and continuous contexts, with significantly increased sample efficiency gains in higher-dimensional BO tasks. An exciting future direction involves discovering $\Phi$ for a broader range of BED tasks such as SIR parameter identification, as well as expanding our approach to misspecified BED tasks.

## Acknowledgments and Disclosure of Funding

This work was supported by the U.S. Army Research Office and the U.S. Army Futures Command under Contract No. W911NF-20-D-0002

We would also like to thank Elissa Wu for her continuous support and encouragement during this project.

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

## A Limitations

This work has several limitations. First, all experiments are conducted on synthetic Bayesian experiment design task instances. While these are widely used benchmarks, they do not capture the full complexity of real-world applications. Second, we assume well-specified models throughout, which may not hold in practice. Finally, our framework presumes access to exact posterior moments (e.g., means and covariances), which is appropriate under fixed priors. However, many practical BED settings involve hyperpriors (such as distributions over kernel lengthscales) that induce posterior distributions over parameters for which first- and second-order moments may no longer be sufficient. This complicates the construction of GNN input representations and poses a challenge for generalizing our approach.

## B Compute Resources

All experiments in this paper were conducted on a cluster of 8 NVIDIA 2080 Ti GPUs. The longest-running experiments involved training the 10 Transformer model seeds on the 8D Bayesian Optimization continuous behavior cloning task, which took approximately one week.

## C Proof of Theorem 1

**Theorem 1** (**Domain Permutation Equivariance**). *If our BED task has the following two properties for all permutations $\phi \in \Phi$:*

1. $r(s, a) = r(\phi(s), \phi(a))$                *(reward invariance)*

2. $p(s'|s, a) = p(\phi(s')|\phi(s), \phi(a))$       *(transition invariance)*

*then, for all permutations $\phi \in \Phi$ for all beliefs $s$ for all discounts $\gamma$, $\pi_\gamma^*(\phi(s)) = \phi(\pi_\gamma^*(s))$ (optimal policy equivariance)*

**Proof:** We'd like to prove that $\pi_\gamma^*(s) = \phi^{-1}(\pi_\gamma^*(\phi(s))) \ \forall s$. This is equivalent to showing that

$$\arg\max_a q^*(s, a) = \phi^{-1}(\arg\max_a q^*(\phi(s), a)) \ \forall s \tag{1}$$

We will also show that

$$q^*(s, a) = q^*(\phi(s), \phi(a))) \ \forall s, \forall a \tag{2}$$

We will prove both Eq 1 and Eq 2 using induction.

Let's start with the last timestep $T$. Eq 2 directly holds from reward equivariance.
Proof for Eq 1 at the last timestep:

$\phi^{-1}(\arg\max_a q_T^*(\phi(s), a)) = \phi^{-1}(\arg\max_a r(\phi(s), a))$
(the action-value function at timestep $T$ is equal to the reward)

$\implies \phi^{-1}(\arg\max_a q_T^*(\phi(s), a)) = \arg\max_a r(\phi(s), \phi(a))$
(using $\arg\max_x f(\phi(x)) = \phi^{-1}(\arg\max_x f(x)) \ \forall f, \phi$)

$\implies \phi^{-1}(\arg\max_a q_T^*(\phi(s), a)) = \arg\max_a r(s, a)$
(using reward invariance)

$\implies \phi^{-1}(\arg\max_a q_T^*(\phi(s), a)) = \arg\max_a q_T^*(s, a).$

Now we'd like to show that if Eq 1 and Eq 2 holds for timestep $t + 1$, it is also true for timestep $t$.

$$\phi^{-1}(\arg\max_a q_t^*(\phi(s), a)) = \phi^{-1}(\arg\max_a r(\phi(s), a) + \gamma\mathbb{E}_{s'|\phi(s),a}[\max_{a'} q_{t+1}^*(s', a')])$$

(Bellman equation)

$$\implies \phi^{-1}(\arg\max_a q_t^*(\phi(s), a)) = \arg\max_a r(\phi(s), \phi(a)) + \gamma\mathbb{E}_{s'|\phi(s),\phi(a)}[\max_{a'} q_{t+1}^*(s', a')]$$

(using $\arg\max_x f(\phi(x)) = \phi^{-1}(\arg\max_x f(x))$)

$$\implies \phi^{-1}(\arg\max_a q_t^*(\phi(s), a)) = \arg\max_a r(s, a) + \gamma\mathbb{E}_{s'|\phi(s),\phi(a)}[\max_{a'} q_{t+1}^*(s', a')]$$

(reward invariance)

$$\implies \phi^{-1}(\arg\max_a q_t^*(\phi(s), a)) = \arg\max_a r(s, a) + \gamma\mathbb{E}_{s'|s,a}[\max_{a'} q_{t+1}^*(\phi(s'), a')]$$

(transition invariance)

$$\implies \phi^{-1}(\arg\max_a q_t^*(\phi(s), a)) = \arg\max_a r(s, a) + \gamma\mathbb{E}_{\phi(s')|\phi(s),\phi(a)}[\max_{a'} q_{t+1}^*(\phi^{-1}(s'), \phi(a'))]$$

(change of variables: $\mathbb{E}_{\phi(x)}[f(x)] = \mathbb{E}_x[f(\phi^{-1}(x))]$)

$$\implies \phi^{-1}(\arg\max_a q_t^*(\phi(s), a)) = \arg\max_a r(s, a) + \gamma\mathbb{E}_{s'|s,a}[\max_{a'} q_{t+1}^*(\phi^{-1}(s'), a')]$$

(transition invariance)

$$\implies \phi^{-1}(\arg\max_a q_t^*(\phi(s), a)) = \arg\max_a r(s, a) + \gamma\mathbb{E}_{s'|s,a}[\max_{a'} q_{t+1}^*(\phi^{-1}(s'), \phi^{-1}(a'))]$$

(invariance of $\max$ under $\phi$)

$$\implies \phi^{-1}(\arg\max_a q_t^*(\phi(s), a)) = \arg\max_a r(s, a) + \gamma\mathbb{E}_{s'|s,a}[\max_{a'} q_{t+1}^*(s', a')]$$

(inductive assumption: Eq. 2 holds for $t + 1$)

$$\implies \phi^{-1}(\arg\max_a q_t^*(\phi(s), a)) = \arg\max_a q_t^*(s, a) = q^*(s, a)$$

This completes our proof by induction. $\qquad\square$

# D   Discrete Belief Representations and Policy Inputs

# E   Further Experiments

## E.1   Full Behavior Cloning Experements

In this section, we present the full results from our behavior cloning experiments described in section 5.2

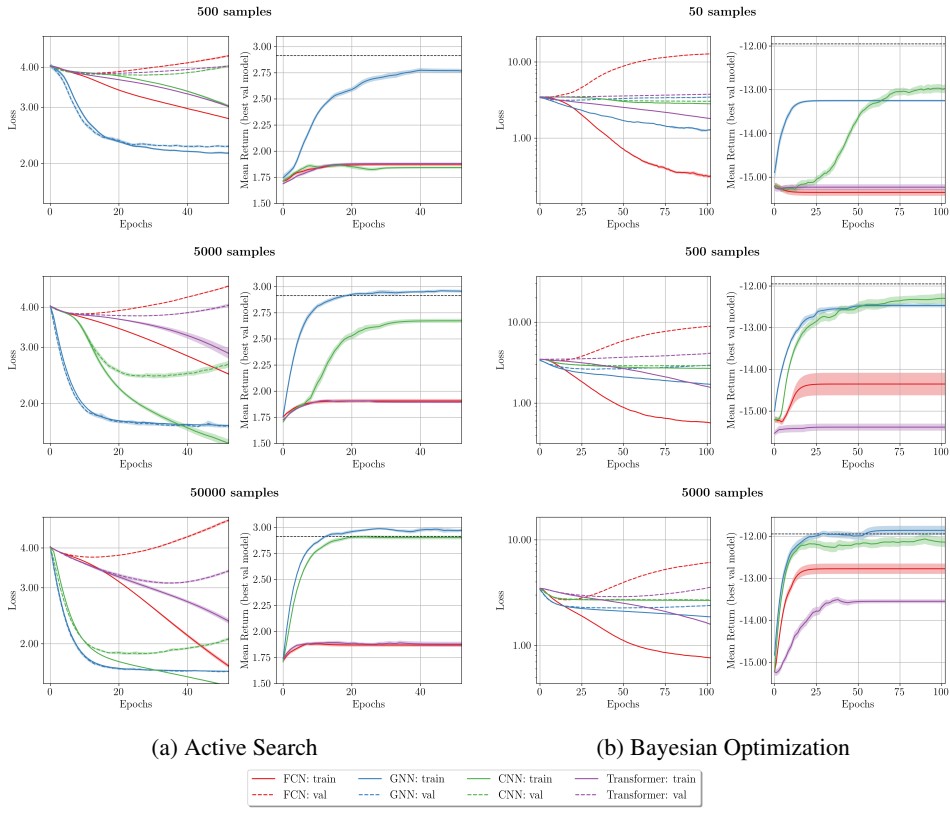

(a) Active Search  (b) Bayesian Optimization

Figure 9: **Comparison of Inductive Biases for Behavior Cloning:** We plot the (i) train (solid lines) and test (dotted lines) behavior-cloning loss and (ii) expected return for various networks on the Active Search (left) and Bayesian Optimization (right) tasks using increasing dataset size (top to bottom). The dotted black line denotes the oracle's expected return. Note that each curve is averaged over 5 seeds. The shaded region shows the 95% confidence interval for the mean.

## E.2    Generalization and Scaling Experiments

In this section, we evaluate whether GNN-based policies can scale to large search domains where established acquisition functions are prohibitively expensive to compute. We show results on three transfer setups in Table 2-4, training the GNN policy on a smaller problem and evaluating transfer performance on larger versions of the problem in each setup.

For example, Table 1, 2 show the performance of a GNN trained using behavior cloning a greedy expert on a $8 \times 8$ active search environment, and evaluated on a $32 \times 32$ environment. The GNN almost matches the performance of the expert on the $8 \times 8$ grid, and allows us to scale to $32 \times 32$ grid. In contrast, it's computationally intractable to run the greedy search on the larger grid because it requires us to estimate the expected information gain for each candidate action, which in turn involves nested integrals. Table 4 and 5 similarly show that GNN-based policies can scale to higher-dimensional tasks and larger grids, and match or outperform expert performance.

Table 1: **Active Search Scaling (Train: $8 \times 8$ grid)**

| Method | Performance | Time (ms) |
|---|---|---|
| Random | $15.28 \pm 0.32$ | $< 0.01$ |
| Simple Search | $12.67 \pm 0.27$ | $< 0.01$ |
| Greedy Search | $\mathbf{35.72 \pm 0.40}$ | $265.4$ |
| GNN | $34.13 \pm 0.49$ | $3.34$ |

Table 2: **Active Search Scaling (Test: $32 \times 32$ grid)**

| Method | Performance | Time (ms) |
|---|---|---|
| Random | $3.72 \pm 0.08$ | $< 0.01$ |
| Simple Search | $3.16 \pm 0.07$ | $< 0.01$ |
| Greedy Search | N.A. | $> 1e4$ |
| GNN | $\mathbf{7.15 \pm 0.12}$ | $67.50$ |

## F    Continuous BO Experiments

Here we present results of experiments comparing the simple regret performance of Transformer and GNN models trained to clone the behavior of the LogEI policy acting on various continuous BO tasks (2D, 4D, 8D), each rollout using $T = 32$ timesteps. In these experiments, no discretization is used: both the Transformer and GNN take the information set $\mathcal{I}_t$ as input and output a point directly in continuous design space, $\mathbf{a}_t \in \mathcal{X} \subseteq \mathbb{R}^d$. This is desirable for contexts where discretization is inappropriate for the level of precision demanded by the application or where it is necessary to additionally amortize inference as well as acquisition estimation and optimization.

We use the BoTorch [41] implementation of LogEI as our expert oracle with default hyperparameters as recommended in their documentation [2]. This oracle is a numerically robust implementation of the canonical Expected Improvement acquisition function, as described in [42]. We use a zero mean Gaussian noise assumption with noise variance of $10^{-6}$.

We train both the Transformer and the GNN models on expert trajectories using standard behaviour cloning losses. Table 3 shows the key architectural details of both models.

Table 3: Transformer and GNN hyperparameters for continuous BO experiments

| Model | Trainable Params | Hidden Dim | # Heads | # Layers |
|---|---|---|---|---|
| Transformer | $\sim 1 \times 10^7$ | 40 | 8 | 10 |
| GNN | $\sim 1 \times 10^7$ | 64 | 8 | 10 |

As in the discrete case, the GNN input is a graph constructed from the information set $\mathcal{I}_t$. Specifically, we use the same basic architecture as in the discrete experiments, but here our graph contains $t - 1$ nodes at timestep $t$ instead of a fixed number of nodes for all timesteps to match a desired level of discretization. The nodal features are the observed values $\{y_\tau\}_{\tau=0}^{t-1}$, and the edge feature for edge $i, j$ is the kernel evaluation $k(x_i, x_j)$. In the discrete experiments, the GNN output logits for the discrete set of actions. This allowed the network to naturally take advantage of domain transformation equivariance. In these continuous experiments, we transform the nodal features to a final set

---

[2]https://botorch.org/

of linear weights $\{w_\tau\}_{\tau=0}^{t-1}$ which are used to combine the points in the information set to define an action distribution for the next timestep:

$$\mathbf{a}_t \sim \mathcal{N}\left( \sum_{\tau=0}^{t-1} w_\tau x_\tau \,,\, \sigma^2 \right) \text{ where } \{w_\tau\} = f_\theta^{\text{GNN}}(\mathcal{I}_t).$$

This parameterization is equivariant to the same set of transformations $\Phi$ that the kernel $k$ is invariant to. Note that in the discrete context, $\Phi \ni \phi : [N] \to [N]$ is a permutation of the discretized domain, while in this continuous setting we abuse notation and use $\Phi \ni \phi : \mathbb{R}^d \to \mathbb{R}^d$ to denote pointwise invertible transformations of the domain. In our experimental results below, we again use an RBF kernel with a lengthscale of 0.2, 0.3, 0.4, for $d = 2, 4, 8$, over an optimization domain refined to the disk $\mathcal{X} = \left\{ x \in \mathbb{R}^d \mid \|x\|_2 \leq 0.5 \right\}$. This results in $\Phi = \mathrm{O}(d)$, where $\mathrm{O}(d)$ denotes the orthogonal group, the group of distance-preserving transformations of $d$-dimensional Euclidean space. We use a fixed variance actor while training both the GNN and Transformer, and during simple-regret rollouts we act deterministically by following the mean policy $\mathbf{a}_t = \sum_{\tau=0}^{t-1} w_\tau x_\tau$. Finally, note that we warm-start the information sets in every rollout with $d + 1$ uniformly at random chosen points in $\mathcal{X}$ for all methods. Timestep 0 in the figure below refers to the first meaningful decision after the warm-start points have already been collected.

The figures below show that GNNs significantly outperform Transformers when trained with the same dataset, with increased delta as BO task dimension increases. The GNN can come close to the simple regret performance of the expert in each task with at least an order of magnitude less data than the Transformer. Moreover, both GNN and Transformer save roughly 1 order of magnitude test-time compute by amortizing the inference, acquisition evaluation and optimization necessary for the BoTorch LogEI expert.

These results demonstrate that our primary equivariance insight is not just restricted to discrete domains. This is important, since real-world BED tasks often involve continuous modeling choices to enable precise experiment designs.

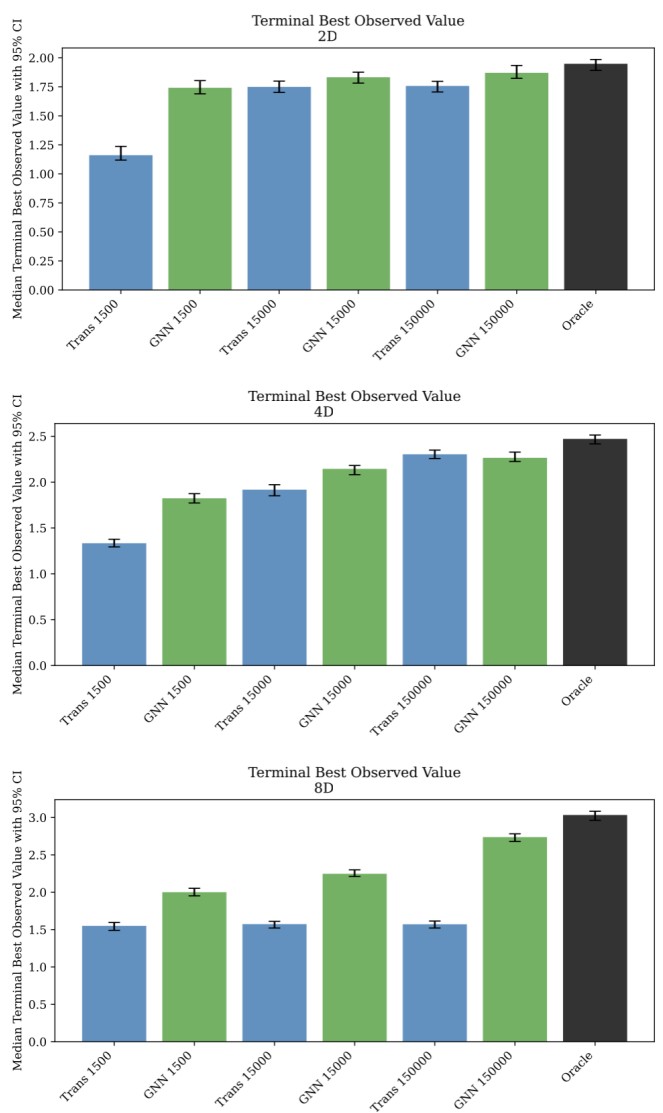

Figure 10: **Continuous BO experimental results.** We use 1,000 rollouts to evaluate the terminal best observed value. Note that optimizing this quantity is equivalent to optimizing simple regret. Here we use median and 1,000-seed bootstrap confidence intervals as is standard in continuous BO literature due to the heavy-tailed nature of simple regret curves in continuous BO, such as in [43]. Both the GNN and Transformer models were trained until overfitting on the indicated number of expert rollout trajectories, with 200 trajectories held out for validation. The simple performance is then computed using the models that achieved the lowest behavior cloning validation loss.

## G   Additional Generalisation Results

In this section, we evaluate the generalization capabilities of our amortized BO policies across search domain dimensions and sizes. Note that these results use discrete belief representations as in the main paper.

Table 4 reports results for a policy trained on a 2D BO task and then evaluated on a set of higher-dimensional BO tasks. We use an adaptively discretized domain inspired by Volpp et al. [7] (explained below) to keep the size of the belief state tractable. The GNN was trained by behavior cloning the Expected Improvement (EI) policy using 5,000 samples, and was evaluated zero-shot on the 3D and 5D tasks. We compare its performance with a random policy, Thompson Sampling

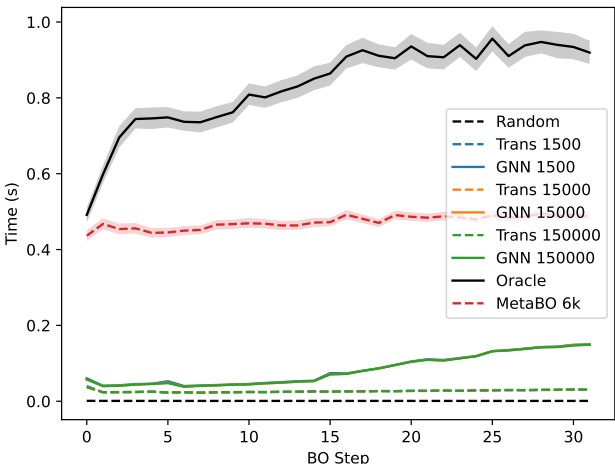

Figure 11: **Continuous BO timing results.** We additionally include a MetaBO variant of the LogEI expert that restricts the acqusition function optimisation to the adaptive discretisatino grid proposed in MetaBO paper [7]. Note that amortized methods significantly outperform non-amortized methods on wall clock time while maintaining comparable performance in terms of simple regret.

(TS), and the EI policy restricted to the discrete domain points. We note that the GNN matches the EI policy's performance on the 2D training task and generalizes well to the 3D and 5D domains.

Table 5 presents another generalization experiment: we train the GNN by behavior cloning the Expected Information Gain (EIG) policy on a domain of $[0, 4]$ discretized into 32 bins using 5,000 samples, and tested on a larger domain of $[0, 128]$ discretized into 1,024 bins. Once again, the GNN shows strong imitation and zero-shot generalization. Note that, due to the computational cost of the EIG policy (exceeding 10 seconds per decision), direct evaluation on the large domain is infeasible. This underscores the value of generalizable amortized policies.

Table 4: **Scaling Amortized BO Policies with Search Dimension**

| Method | Train: 2D | | Test: 3D | | Test: 5D | |
| | Performance | Time (ms) | Performance | Time (ms) | Performance | Time (ms) |
| --- | --- | --- | --- | --- | --- | --- |
| Random | $1.07 \pm 0.05$ | 0.02 | $1.13 \pm 0.05$ | 0.02 | $1.13 \pm 0.05$ | 0.02 |
| TS | $1.25 \pm 0.05$ | 1.13 | $1.29 \pm 0.05$ | 1.86 | $1.23 \pm 0.05$ | 1.04 |
| EI | $\mathbf{1.31 \pm 0.05}$ | 0.29 | $\mathbf{1.48 \pm 0.06}$ | 0.30 | $\mathbf{1.48 \pm 0.05}$ | 1.94 |
| GNN | $\mathbf{1.28 \pm 0.05}$ | 67.01 | $\mathbf{1.49 \pm 0.06}$ | 56.32 | $\mathbf{1.47 \pm 0.05}$ | 57.46 |

Table 5: **Scaling Amortized BO Policies with Grid Size**

| Method | Train: $N = 32$ | | Test: $N = 1024$ | |
| | Performance | Time (ms) | Performance | Time (ms) |
| --- | --- | --- | --- | --- |
| Random | $-15.43 \pm 0.17$ | 0.05 | $-33.79 \pm 0.27$ | $< 0.11$ |
| TS | $-13.44 \pm 0.20$ | 0.21 | $-29.21 \pm 0.38$ | 29.11 |
| UCB | $-13.74 \pm 0.17$ | 0.15 | $-29.10 \pm 0.33$ | 0.24 |
| EIG | $\mathbf{-11.95 \pm 0.20}$ | 1067.4 | N.A. | $> 1e4$ |
| GNN | $\mathbf{-11.60 \pm 0.20}$ | 9.83 | $\mathbf{-27.94 \pm 0.36}$ | 70.63 |

Adaptive discretization method: Sample $n$ points uniformly at random in the domain, choose the top $k$ using the EI acquisition function, and then sample $m$ points near each of these $k$ best points. The candidate set consists of $n + km$ points. We use $n = 100, k = 5, m = 20$.

Evaluation procedure: For all the experiments in this section, we evaluate the mean return of the policy using a fixed episode length of 8, averaged over 200 trials, using the EI reward for Table 4 and EIG reward for Table 5. Since the EI reward requires a previous max-value, instead of assuming

a fixed initial max value, we choose to initialize each trial with a single point chosen uniformly at random in the domain. This point does not contribute to the timestep count in our experiments.

## H  Generalization through data augmentation

In this experiment, we compare two different strategies for leveraging the domain permutation equivariant structure in Bayesian Optimization policies for efficient behavior-cloning: (i) using a network with a suitable inductive bias, i.e., the Graph Neural Network described in Section 4, and (ii) applying data augmentation during training. This data augmentation corresponds to the set of permutations the policy is equivariant to.

Figure 12 shows the behavior-cloning training and validation loss alongside the resulting policy performance across several network architectures. Additionally, it shows outcomes from training a fully connected network (FCN) using data augmentation. The results indicate that data augmentation significantly improves the generalization capability of the FCN, as evidenced by the near-perfect alignment of training and validation loss curves (depicted in purple). However, this improvement comes at a substantial computational cost—training times are considerably prolonged. Notably, the GNN achieves expert-level performance within approximately 30 epochs, whereas the FCN augmented with permutation-based data augmentation fails to match this performance even after 7,000 epochs. This disparity is expected since the equivariant permutation set is very large, equal to $n!$ for a $n$-bin discrete BO task. This is the number of permutation the GNN is naturally equivariant to, but the FCN needs explicit data augmentation for during training. For this particular experiment, the number of permutations is $n! = 32! \approx 2.6 \times 10^{35}$.

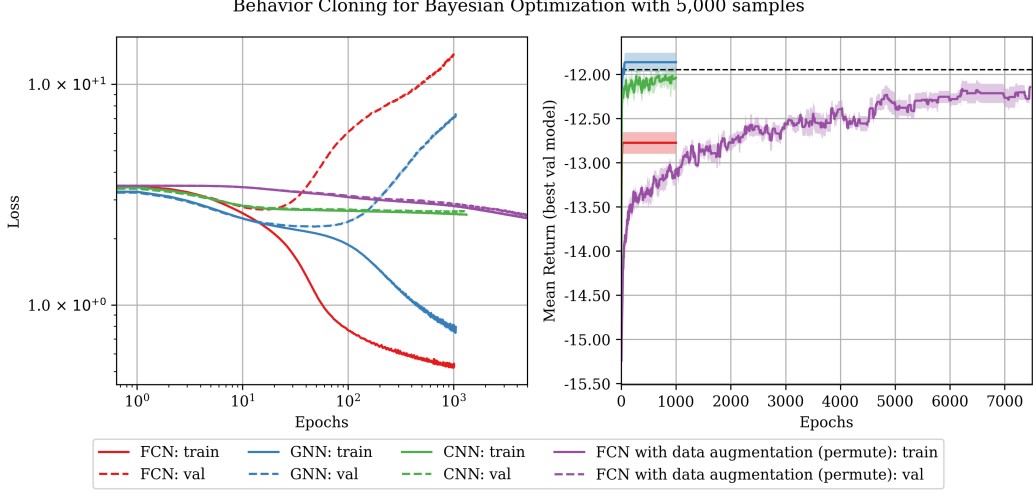

Figure 12: **Comparison of Inductive Biases and Data Augmentation for Behavior Cloning:** We plot the (i) train (solid lines) and test (dotted lines) behavior-cloning loss and (ii) expected return for various networks on the Bayesian Optimization using a dataset size of 5,000 samples. The oracle's expected return is indicated by the dotted black line. Regular training was terminated at 1,000 epochs after performance had stagnated. Note that each curve is averaged over 5 seeds. The shaded region shows the 95% confidence interval for the mean.

## I  Generalization and Scalability for Bayesian Optimization

In this section, we discuss the generalizability and scalability of our amortized approach to Bayesian Optimization (BO).

### I.1  Generalizability

In all but the scaling experiments of section H, our experimental setup assumes a well-specified prior: both training and test functions are sampled from a known Gaussian Process (GP) prior.

While this assumption facilitates controlled evaluation, it does not always hold in practice. Real-world applications of BO often involve misspecified priors, where the true function does not match the assumed GP prior. In these cases, surrogate models are fit by estimating GP hyperparameters from data.

To generalize our amortization approach beyond the well-specified regime, the learned policy should (i) train on a distribution of BED tasks induced by our prior distribution over hyperparameters, and (ii) condition on the posterior distribution over the full set of random variables in the BED task, which now includes the hyperparameters.

## I.2 Scalability

Our approach can help with scalability by amortizing the computationally expensive BO methods via a neural policy that maps belief states to actions, thereby reducing online computational burden. However, special care is required in representing the belief state. Our method relies on an explicit representation of the posterior over the search space, which can become prohibitively expensive in high-dimensional settings. In particular, a naïve uniform discretization leads to a grid whose size grows exponentially with the number of input dimensions.

To mitigate this, we adopted a simple adaptive discretization strategy inspired by MetaBO [7] for our higher dimensional BO experiments in section G. By concentrating representational capacity in high-mean or high-uncertainty regions, we retain a tractable belief representation. Nonetheless, the choice of discretization remains a critical factor, and a more principled exploration of this design choice is left to future work.

## J  Network Hyperparamters for Discrete Experiments

Table 6: **Hyperparameters for FCN.**

| Hidden Layers | [512, 512, 512] |
|---|---|

Table 7: **Hyperparameters for CNN.**

| Number of Layers | 6 |
|---|---|
| Kernel size | 9 |

Table 8: **Hyperparameters for GNN.**

| Hidden Dim | 64 |
|---|---|
| Number of Heads | 8 |
| Number of Layers | 2 |

Table 9: **Hyperparameters for Transformer.**

| Number of Layers | 5 |
|---|---|
| Number of Heads | 16 |
| Embedding Size | 64 |

# K   Training Hyperparamters

Table 10: **Behavior Cloning Hyperparameters**

| Learning Rate | 3e-4 |
|---|---|
| Batch Size | 32 |
| Optimizer | Adam |

Table 11: **Reinforcement Learning Hyperparameters**

| Algorithm | DDQN |
|---|---|
| Exploration | $\epsilon$-greedy |
| Discount factor | 0.95 |
| Learning Rate | 3e-4 |
| Batch Size | 32 |
| Optimizer | Adam |

We also use non-stationary Q-functions with DDQN since we model our BED tasks as finite-horizon MDPs. In addition, we use integrated Bellman targets to reduce noise in training.

# L   Visualisation of Evolution of GNN & FCN Critic

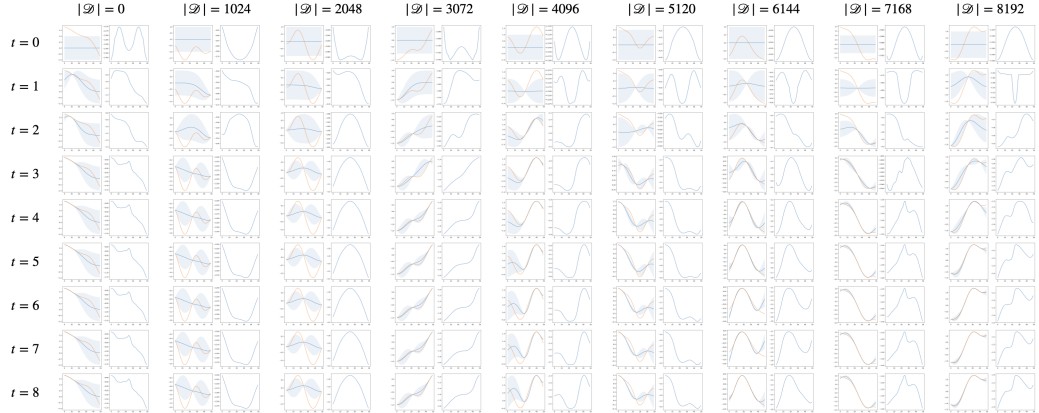

Figure 13: **Evolution of BO GNN policy through RL training.** Each column illustrates a rollout after a given number of collected MDP transitions (indicated in the column title). The left figure in each column depicts the belief state $p_t$ and ground-truth hidden function, and the right figure illustrates the estimated $q$-values. Note that the GNN quickly learns myopic search behavior (first check both corners of the domain). After gathering more experience, it instead learns to first query the internal point of the domain to act less greedy while maximizing return. Note also that the GNN from initialisation captures symmetry and smoothness priors, and that after 8,192 transitions has learned to represent both sharp discontinuities and smoothness, which is essential when modeling the drop in value for re-sensing a previously queried point, but sharp increase in value for querying neighboring points if the observed point is high.

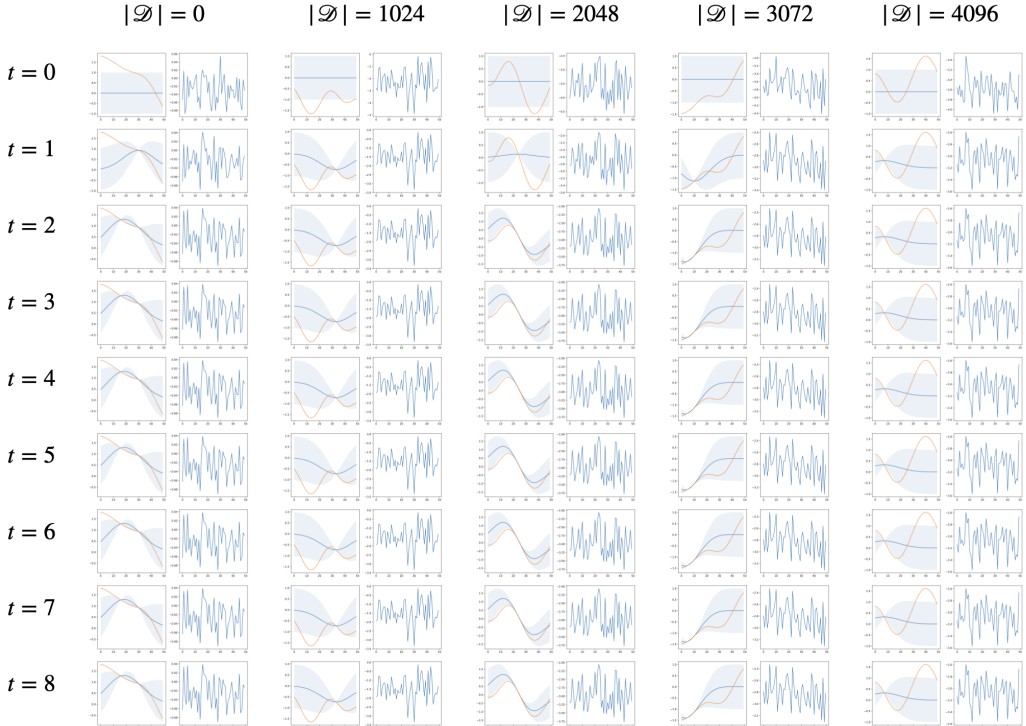

| | $|\mathcal{D}| = 0$ | $|\mathcal{D}| = 1024$ | $|\mathcal{D}| = 2048$ | $|\mathcal{D}| = 3072$ | $|\mathcal{D}| = 4096$ |

Figure 14: **Evolution of BO FCN policy through RL training.** Each column illustrates a rollout after a given number of collected MDP transitions. The left figure in each column depicts the belief state $p_t$, and the right figure illustrates the estimated $q$-values. Note that no qualitative improvement to the q-values is visible after 4,096 MDP transitions, in contrast to the GNN which achieves high rewards and clear structure in the $q$-estimates.

