# OpenReview forum: "Efficient Bayesian Experiment Design with Equivariant Networks"
_NeurIPS.cc/2025/Conference — NeurIPS 2025 poster_

### Official Review · Reviewer_mCGY · 2025-06-27

**Clarity:** 3
**Significance:** 3
**Originality:** 3
**Rating:** 5
**Confidence:** 3

**Summary:**

The authors prove that many Bayesian Experimental Design (BED) problems are domain permutation invariant. To exploit this property, they propose to use Graph Neural Networks (GNNs), which are inherently permutation equivariant. Through extensive experiments, they show that leveraging this architectural bias can dramatically improve sampling efficiency, unlocking significantly larger tasks, making progress towards addressing the "belief explosion" issue.

**Questions:**

- The limitations section points out that handling hyperpriors is an open challenge. Can you elaborate on how the framework might be extended to handle this?
- For the continuous experiments, the GNN outputs weights to form a linear combination of points. Are there cases where this strategy falls short?
- As acknowledged in the limitations section, in real-world applications, model misspecification will be common. How robust is the method to model misspecification? What might be ways to improve robustness?

**Ethical Concerns:**

["NO or VERY MINOR ethics concerns only"]

**Final Justification:**

I appreciated the thorough response of the authors, which resolved my questions, and maintain my recommendation to accept the paper.

**Limitations:**

Yes

**Paper Formatting Concerns:**

No major issues

**Quality:**

3

**Strengths And Weaknesses:**

## Strengths

- Novel, original approach towards better solutions on relevant problems
- Sound theoretical foundation
- Comprehensive quantitative evaluation
- Paper is clearly written, good presentation of results

## Weaknesses

- Rests on the assumption of well-specified models, which is commonly violated in practice
- Reliance on first- and second-order posterior moments; insufficient for handling hyperpriors (as discussed by the authors)

### Minor

- Readability could be improved by placing citations in brackets

---

> ### Author Rebuttal · Authors · 2025-07-31
>
> We would like to thank the reviewer for their time and insightful questions. We appreciate the positive comments, and we respond to the individual questions below.
>
> **For the continuous experiments, the GNN outputs weights to form a linear combination of points. Are there cases where this strategy falls short?**
>
> Yes, this strategy can fall short if the information set does not sufficiently span the design space. Specifically, if the information set contains points that lie within a low-dimensional subspace of $\mathcal{X}$, then any linear combination of those points will remain within that subspace, limiting the exploration capabilities of the policy.
>
> To address this, we initialize each trial with $d+1$ points sampled uniformly at random from $\mathcal{X}$, where $d$ is the dimensionality of the design space. This ensures (almost surely) that the information set spans $\mathcal{X}$. This warm-start practice is common in continuous Bayesian optimization and aligns with protocols in the literature. For example, Astudillo & Frazier (2019) write: “For all problems and methods, an initial stage of evaluations is performed using $2(d + 1)$ points chosen uniformly at random over $\mathcal{X}$.” [1]
>
> Thus, while the linear combination strategy has inherent limitations, we mitigate them through principled initialization that ensures sufficient coverage of the design space.
>
> [1] Astudillo, Raul, and Peter Frazier. "Bayesian optimization of composite functions." International Conference on Machine Learning, 2019.
>
> **The limitations section points out that handling hyperpriors is an open challenge. Can you elaborate on how the framework might be extended to handle this?**
>
> In the continuous setting, one natural extension is to train the GNN on information sets generated using a distribution over hyperparameters e.g., sampling kernels or other GP hyperparameters at training time. This effectively averages over a richer space of data-generating processes, helping the model learn representations that are robust to different prior assumptions.
> For the discrete setting, a viable alternative is to use MAP estimates of kernel hyperparameters (obtained via standard hyperprior inference) to compute GP posterior moments. These moments can then be encoded as input graphs, as done in our current framework. While this collapses the full Bayesian belief state into a second-order approximation, we hypothesize that it suffices for many realistic forms of misspecification.
>
> Nonetheless, we agree this is an important and nuanced direction, and a dedicated study of hyperprior graph representation is warranted.
>
> **As acknowledged in the limitations section, in real-world applications, model misspecification will be common. How robust is the method to model misspecification? What might be ways to improve robustness?**
>
> We appreciate the reviewer’s emphasis on this important issue. In general, amortized policies (such as those learned by our GNN) are inherently sensitive to discrepancies between training-time and test-time data-generating processes. That said, robustness can potentially be improved by diversifying the training data.
>
> For instance, if even a few representative real-world functions are available, one could augment the training set with these alongside synthetic GP samples. This could enable the policy to implicitly learn priors over a broader and potentially non-Gaussian class of functions. Such a strategy would allow the policy to adapt beyond the GP framework without requiring explicit changes to the architecture or the inference mechanism.
>
> Finally, we believe that there are many competing reasonable notions of misspecification and robustness that warrant careful study. We agree that a dedicated study of robustness to model mispecification is an important future direction of this work.

---

> > ### Comment · Reviewer_mCGY · 2025-08-05
> >
> > Thank you for the thoughtful replies; I will keep my score (accept).

---

### Official Review · Reviewer_Hk2o · 2025-06-29

**Clarity:** 3
**Significance:** 3
**Originality:** 2
**Rating:** 4
**Confidence:** 4

**Summary:**

This paper discusses the (modelling) choice of a (deep) policy network for Bayesian Experiment Design (BED) tasks (i.e., a generalization of Bayesian Optimization and/or Active Search). It argues that the belief state undergoes an exponential explosion with as the time horizon increases (a well known result since the 1960's which led to the term "curse of dimensionality" by Bellmann). The paper argues that in BED tasks which have a state-permutation invariance in both the MDP model and reward function, a Graph Neural Network architecture provides the right inductive bias to reduce the number of necessary samples as well as avoids overfitting. The claim is empirically validated on a toy problem.

**Questions:**

How exactly is the state space mapped to the nodes of a GNN and CNN for both the Bayesian Optimization and Active Search example in Section 5?

**Ethical Concerns:**

["NO or VERY MINOR ethics concerns only"]

**Final Justification:**

Thanks for the clarification by the authors in the rebuttal - I have adjusted my scores accordingly

**Limitations:**

yes

**Paper Formatting Concerns:**

* In general, the citation style is inconsistent when a citation occurs as part of the sentence.
* Line 34: BO (Bayesian Optimization) is not introduced.
* Line 125: Hasn't this problem already been observed in the original Bellmann paper?
* Line 178: The notation $\mathbb{S}$ has not been introduced.
* Line 230: The notation $\mathcal{N}$ has not been introduced.
* Line 233 & 235: $X_{\mathrm{disc}}$ is not consistently used with roman font for the subscript.
* Line 243: I find the index $i$ in the $x_i$ unfortunate as $i$ runs over random draws of $f$. Maybe simply used $x$?

**Quality:**

2

**Strengths And Weaknesses:**

A strength of the paper is the empirical evaluation on a rather small, discrete state-space example which already highlights the substantial differences in generalization performance - also compared to non DNN approaches such as UCB or Thompson sampling. For me, a major weakness is the vague description how the neural network architectures are mapped to the policy network; I still find it hard to understand how exactly the GNN is mapped to this problem to achieve the improved generalization. As this is the main contribution of the paper, I would have preferred more clarity there (Figure 4 is too vague for my taste)

---

> ### Author Rebuttal · Authors · 2025-07-30
>
> We would like to thank the reviewer for their time and their valuable feedback. We appreciate the positive comments on our empirical evaluation. We would like to mention that we organized the paper to contain the most pedagogically understandable examples in the main paper but include significantly more experiments in the appendix, including fully continuous higher-dimensional (4D and 8D) Bayesian Optimization tasks.
>
> The reviewer is right to point out that the description of how the GNN is mapped to the policy network could be clearer, and we are happy to provide a more detailed description below.
>
> As a reminder of our setup, our policy network maps from the belief-state to a distribution over actions, and our goal is design a domain-permutation equivariant policy. For example, in Bayesian Optimization, this belief-state corresponds to the mean vector $\mu$ and covariance matrix $\Sigma$, and the simplest example of a domain permutation we want the policy to be equivariant to is flips for 1D problems (generalized to rotations for higher-dimensional BO problems).
>
> Our parameterization of the input graph to the GNN is directly motivated by these domain-permutation equivariances. Each node of the graph corresponds to a point in the domain; the node features are the marginal mean and variance, and the edge features are the covariance terms. The GNN processes this input graph and the final layer outputs a single logit for each node in the graph. Since each node corresponds to a point in the domain, this gives us a logit for each possible discrete action. We then apply a softmax function over these logits to produce the final probability distribution for our policy.
>
> This specific parameterization ensures that the desired equivariance is built into the architecture. Any permutation of the graph's nodes (e.g., a rotation or flip of the domain) will result in an identical permutation of the output logits, thereby making the policy exactly equivariant. This structure allows the network to generalize far more efficiently, as it learns a single underlying mapping to action distributions that applies across all symmetric belief states.
>
> We describe the design for a similarly motivated input graph for the GNN for fully-continuous BO tasks in Section E of the appendix.
>
> We thank the reviewer again for this constructive feedback, and we will update our manuscript to include a detailed step-by-step description of the GNN.
>
> With regards to the Bellman curse of dimensionality observation, we agree with the reviewer that our observation about belief explosion is strongly related to Bellman’s original observation. We acknowledge this when introducing the concept of belief explosion in our paper on line 125 by connecting it to the literature on POMDP planning. However, while Bellman’s original observation concerned a broad class of decision-making under uncertainty problems, part of our contribution is to highlight belief explosion for BED problems specifically and how existing contemporary approaches fail to recognise this as a training bottleneck. Moreover, our central contribution is to **propose a specific solution for certain BED problems** which tames this problem significantly.
>
> We thank the reviewer for the formatting and notation issues, we will address all of these issues in the final manuscript.

---

> > ### Comment · Reviewer_Hk2o · 2025-08-07
> >
> > Thanks for the clarification - this helped a great deal! I have adjusted my scores.

---

### Official Review · Reviewer_Dbo7 · 2025-07-03

**Clarity:** 3
**Significance:** 2
**Originality:** 2
**Rating:** 4
**Confidence:** 2

**Summary:**

This paper tackles Bayesian Experiment Design (BED) with deep learning by mitigating the belief explosion issue (similar to low sample efficiency issue in reinforcement learning). The authors propose to leverage an appropriate inductive bias for the deep neural networks as a solution. Specifically, they adopt graph neural network due to their domain permutation equivariance properties. Their experiments show that both this design outperform other alternatives (CNN, Transformer, etc.) on two instances of BED tasks, namely Bayesian Optimization and Active Search.

**Questions:**

See "weaknesses"

**Ethical Concerns:**

["NO or VERY MINOR ethics concerns only"]

**Final Justification:**

I remain my original rating, i.e, learning towards acceptance.

**Limitations:**

yes

**Quality:**

3

**Strengths And Weaknesses:**

Strengths:
+ The paper is well-motivated and explain the belief explosion issue relatively well, which seems to be overlooked in the literature (as what the authors suggest).
+ The experiments show adequate evidence as it draws comparison with several learning-based baselines as well as non-learning-based ones (e.g., Thompson Sampling).

Weaknesses:
+ While GNN is reasonable here, there might be finer-grained network design choices to be studied. I wonder within a generalized class of GNN (or say permutation-equivariant functions), is there specific designs that are better than others. For instance, network structures in other relevant fields has been studied before such as PointNet, Deep Sets, and Set Transformer.

---

> ### Author Rebuttal · Authors · 2025-07-30
>
> We would like to thank the reviewer for their time and for raising this point about our specific choice of GNNs.
>
> We agree with the reviewer that that a study of finer-grained network design choices would be valuable, and that our work suggests GNNs as just one way to learn domain permutation-equivariant functions. However, we would like to point out that while other existing networks like PointNet, Deep Sets and Set Transformers do handle permutation symmetries, they are not suitable to leverage the restricted class of domain permutation-equivariances in BED problems. This distinction arises from the **type of input representation** and the **corresponding type of symmetry** we can leverage. There are two primary ways to formulate a BED policy:
>
> **Option A: Map directly from the** **information set to the action distribution $a_t \sim \pi(\{(x_{\tau}, y_{\tau})\}_{\tau=0}^{t-1})$.** In this setup, the policy is invariant to all permutations of the information set (i.e. it does not matter whether we observe the sequence $(x_a,y_a), (x_b,y_b), (x_c,y_c)$ or $(x_b,y_b), (x_c,y_c), (x_a,y_a)$ , the next optimal action is the same). Any network designed to learn functions over such sets is appropriate, including PointNet, Deep Sets and Set Transformers. Our Transformer baseline (with no positional encoding mechanism) is an example of such a model, and as our main results show, only leveraging information set permutation invariance is insufficient to combat the belief explosion issues we observe.
>
> **Option B: Perform Bayesian inference to compute the belief-state and then** **map from the belief state to the action distribution $a_t \sim \pi(p(\theta|\mathcal{I}_t))$.** This two-step process automatically handles the information-set permutation invariance from the first setup (since inference results in the same posterior regardless of data order). However, it also introduces a new, more structured symmetry: **domain permutation equivariance**. Figure 1 in the paper illustrates this concretely: if we rotate the belief-state, the optimal action distribution should also rotate. GNNs are especially suitable for this task because the belief state (i.e., the posterior mean and covariance) has a natural graph structure as it basically models a set of random variable with marginal and joint statistics. PointNet or Deep Sets would treat the the marginal mean and variances as an unordered set of values, completely ignoring the rich relational information encoded in the **joint** statistics such as the covariance matrix. GNNs, on the other hand, can preserve the relationship between points in the domain by encoding the joint statistics (covariance information) in the graph edges, while being equivariant to domain permutations such as rotations.
>
> We thank the reviewer again for prompting this important discussion. We will add a paragraph to Section 4 to explicitly clarify this distinction and better contextualize our choice of GNNs with respect to these other architectures.

---

### Official Review · Reviewer_sQcj · 2025-07-04

**Clarity:** 3
**Significance:** 2
**Originality:** 2
**Rating:** 4
**Confidence:** 2

**Summary:**

This paper studies Bayesian Experiment Design (BED) and proposes a way to reduce the sample requirements when using deep learning for BED. Specifically, it proposes to leverage graph neural networks to exploit problems that exhibit permutation equivariance in the reward and transition. By taking advantage of the equivariance in the belief state, GNNs generalize better over permuted belief states.

The experiments study the Bayesian Optimization and Active Search tasks, and compare the efficacy of the GNN architecture compared to fully connected networks, convolutional neural networks, and transformers. They also compare to more traditional baselines for BO and AS, including UCB and Thompson Sampling. The results show that the GNNs generalize better than other deep learning architectures, and match the oracle better than the non-learning baselines.

**Questions:**

* How does this differ from prior works that use GNNs to develop deep reinforcement learning algorithms? Does your implementation make specific assumptions for the BED case? For example:
    * Almasen et al. Deep Reinforcement Learning meets Graph Neural Networks: exploring a routing optimization use case.
    * Gammelli et al. Graph Neural Network Reinforcement Learning for Autonomous Mobility-on-Demand Systems.
* To address the issue described in Lines 154-155, i.e., that the network underfits for shallow beliefs but overfits for deep beliefs, would it make sense to have a different network/policy for each time-step?
* Could we see what Fig. 3 (right) would look like when using GNNs?
* In Fig. 6, are the “Mean Return (best val model)” plots showing the best validation return seen so far? How is the validation set defined? I’m also curious about the return achieved by the model at that point in training. Could we see that plot? That would show how much fluctuation in training there is.
* In Fig. 6, do the vastly different losses just indicate that training parameters need to be tuned and that some basic regularization needs to be applied?
* In Fig. 7, what is terminal posterior argmax entropy? Is that the same as the one-step reward or is it the reward at the final time-step? What would the return look like?
* What are the error bars in the figures computed over?

**Ethical Concerns:**

["NO or VERY MINOR ethics concerns only"]

**Final Justification:**

I've read the authors' responses and maintain my recommendation to accept.

**Limitations:**

Yes

**Quality:**

3

**Strengths And Weaknesses:**

Strengths:
* The paper focuses on the problem of Bayesian optimal experimental design, and investigates the use of graph neural networks to reduce the sample requirements that traditional methods require.
* Although the experiments only study simple toy tasks, they seem thorough and well-designed, and do the intended job of showing that the use of GNNs improves the sample efficiency over baselines for different BED tasks.

Weaknesses:
* The introduction helps provide context into BED and the prior approaches to this problem, covering both deep learning and non-deep learning techniques. However, it doesn’t cover GNNs or their prior applications into RL/other sequential decision-making problems. How do those prior works and the conclusions from them relate to this paper?
* Nit: I think more effort into putting the major experiments in the main paper would improve the readability of the paper. Currently, only one preliminary experiment (behavior cloning) is included in the main paper while the more central experiments are in the appendices.

---

> ### Author Rebuttal · Authors · 2025-07-30
>
> We would like to thank the reviewer for their time and valuable feedback. We agree with the reviewer on both of their main points of feedback:
>
> I. As the reviewer points out, GNNs have been commonly used as policy representation to naturally capture the relational structure among components in a system, such as joints in a robot [1] or agents in a multi-agent environment [2], and the paper would benefit from describing this prior use of GNNs and comparing it to our work. The main difference is that prior work on GNN-based policies are motivated by their adaptivity to the size of the input graph and structure (corresponding to varying robot topology and number of agents in these examples). In contrast, our work uses GNNs to leverage policy equivariance that we show exist in (any fixed-sized) BED problems specifically.
>
> 1. Almasan et al., "Deep reinforcement learning meets graph neural networks: Exploring a routing optimization use case.”
> 2. Nurbek et al., "Exploring Graph Neural Networks in Reinforcement Learning: A Comparative Study on Architectures for Locomotion Tasks.”
>
> II. We also agree that readability would improve by moving some continuous and high-dimensional BO experimental results from the appendix to the main body of the paper. To accommodate this, we will condense the behavior cloning experiments by presenting plots for only one representative dataset size per problem type in the main text, shifting additional dataset size results to the appendix. This will allow the reader a greater opportunity to appreciate the range of experiments included in this work.
>
> ---
>
> Answers to reviewer’s questions:
>
> **Q. How does this differ from prior works that use GNNs to develop deep reinforcement learning algorithms? Does your implementation make specific assumptions for the BED case?**
>
> We have provided an answer to this in our response above, but to summarize concisely: while prior works often use GNNs for problems with variable graph structures, our key insight is using a GNN to exploit the domain-permutation equivariance inherent in BED problems. This allows our policy to generalize efficiently across a vast space of possible beliefs, which is an assumption specific to the BED case that we leverage.
>
> **Q. To address the issue described in Lines 154-155, i.e., that the network underfits for shallow beliefs but overfits for deep beliefs, would it make sense to have a different network/policy for each time-step?**
>
> That is an interesting suggestion. However, using a different policy for each timestep would not address the root of the problem. The main bottleneck is that the learning problem becomes fundamentally harder at later timesteps due to the "belief explosion"—the much wider diversity of belief-states the policy encounters.
>
> While it is not the focus of our work, we also expect a single policy network to have some inter-timestep generalization which we would lose by having a different policy network for each timestep.
>
> A related brute-force approach might be to collect significantly more training data for these later timesteps. In contrast, our paper proposes a more sample-efficient solution.
>
> **Q. Could we see what Fig. 3 (right) would look like when using GNNs?**
>
> We observed that the GNN clearly generalizes well across all the nine timesteps in that example. We didn’t include in our original submission due to space constraints and are not able to share it in our response due the NeurIPS rebuttal policy, but we will include it in our final manuscript in the appendix.
>
> **Q. In Fig. 6, are the “Mean Return (best val model)” plots showing the best validation return seen so far? How is the validation set defined? I’m also curious about the return achieved by the model at that point in training. Could we see that plot? That would show how much fluctuation in training there is.**
>
> The validation set consists of a 20% held-out portion of the offline dataset of expert trajectories.
>
> The plot shows the mean return of the policy checkpoint that has achieved the lowest behavior-cloning validation loss up to that epoch.
>
> Rationale: The return of the latest policy has a U-shape, improving initially and then degrading as the policy overfits on the behavior cloning data. We chose this presentation to more clearly illustrate the best achievable online performance for each architecture as a function of training data, factoring out the noise from overfitting. We can add the plots of the return achieved by the model at that point in training to the appendix for completeness.
>
> **Q. In Fig. 6, do the vastly different losses just indicate that training parameters need to be tuned and that some basic regularization needs to be applied?**
>
> Our goal in this experiment was to isolate and compare the effectiveness of the inherent inductive bias of each architecture. Therefore, we intentionally evaluated the models on an equal footing, using a similar number of parameters and without architecture-specific regularization. We chose the simplest parameterization of each network type, choosing the default GNN implementation in the pytorch-geometric library for example, and did not perform any hyperparameter tuning, apart from making sure that all the networks had a similar number of parameters. We will include our motivation for the policy network parameterizations in the appendix.
> Finally, we note that all the policy networks we used were capable of overfitting in these experiments and achieve similar training loss; their validation loss and return vary vastly primarily because of their different generalization ability.
>
> **Q. In Fig. 7, what is terminal posterior argmax entropy? Is that the same as the one-step reward or is it the reward at the final time-step? What would the return look like?**
>
> The "terminal posterior argmax entropy" is the entropy of the belief over the function's maximum location at the final timestep of the episode. In our BO experiments, the per-step reward is the expected reduction in this entropy, also known as Expected Information Gain. This is the same objective that popular entropy-search methods such as PES [4] optimizes.
>
> Maximizing the cumulative return is equivalent to minimizing this terminal entropy, i.e., over the entire trial, not just the next timestep. The bar plot in Figure 7 shows this terminal entropy.
>
> [3] José Miguel Hernández-Lobato et. al. Predictive Entropy Search for Efficient Global Optimization of Black-box Functions
>
> **Q. What are the error bars in the figures computed over?**
>
> The error bars in all training curve plots represent ±2 standard errors of the mean, computed over 5-10 independent runs with different random seeds. We used 5 seeds for the discrete BO and AS experiments, and used 10 seeds for the continuous BO experiments. We can include links to the complete comet training logs for all our experiments with the final manuscript.
>
> For policy evaluation in the continuous BO task, they represent 95% confidence intervals estimated with a hierarchical bootstrap, a standard practice for that domain. Specifically, there are two sources of randomness for the continuous BO Behavior Cloning task: the 10 seeds of model training, and the 1,000 seeds of model evaluation. We use a hierarchical bootstrap estimate of the 95% CI for the statistics we report in the figures in the appendix.

---

> > ### Comment · Reviewer_sQcj · 2025-08-08
> >
> > Thank you for the clarification! I think revising the manuscript with some of these responses (i.e., updating the related work, explanation of the validation set, error bars, and "terminal posterior argmax entropy") will help improve the clarity of the paper.
> >
> > Regarding hyperparameters for the different architectures, I still believe doing some basic tuning via a simple grid search that's comparable for all architectures would strengthen the argument while still maintaining "equal footing."

---

### Decision · Program_Chairs · 2025-09-17

**Decision:**

Accept (poster)

**Comment:**

This paper studies Bayesian Experiment Design and addresses the “belief explosion” problem, where the number of realizable beliefs grows exponentially with experiment length, creating severe sample inefficiency. The authors propose using Graph Neural Networks to exploit the domain permutation equivariance in BED problems, enabling policies that generalize efficiently across symmetric belief states. Experiments on Bayesian Optimization and Active Search show that GNN-based policies outperform alternative architectures (MLPs, CNNs, Transformers) and match or exceed traditional baselines like UCB and Thompson Sampling.

The paper is well-motivated, technically sound, and clearly written, with a novel use of GNNs that is distinct from prior RL applications. Reviewers found the empirical evaluation convincing, and the authors’ rebuttal addressed concerns about architectural design, validation procedures, error bars, and the mapping of belief states to GNN nodes. Limitations include reliance on model assumptions, sensitivity to hyperpriors, and evaluation mostly on toy tasks, but these are acknowledged and discussed.

Overall, the paper makes a solid contribution by combining a principled architectural choice with empirical evidence of improved sample efficiency. Given the technical quality, originality, and thorough responses during the rebuttal, the recommendation is to accept.